# Photocatalytic Activity of TiO_2_ for the Degradation of Anticancer Drugs

**DOI:** 10.3390/nano12193532

**Published:** 2022-10-09

**Authors:** Kristina Tolić Čop, Dragana Mutavdžić Pavlović, Tatjana Gazivoda Kraljević

**Affiliations:** 1Department of Analytical Chemistry, Faculty of Chemical Engineering and Technology, University of Zagreb, Marulićev trg 19, 10000 Zagreb, Croatia; 2Department of Organic Chemistry, Faculty of Chemical Engineering and Technology, University of Zagreb, Marulićev trg 19, 10000 Zagreb, Croatia

**Keywords:** imatinib, crizotinib, photocatalysis, degradation products, water matrix, scavenger study, toxicity

## Abstract

To prevent water pollution, photocatalysis is often used to remove small molecules such as drugs by generating reactive species. This study aimed to determine the photocatalytic activity of two anticancer drugs, imatinib and crizotinib, and to investigate various influences that may alter the kinetic degradation rate and ultimately the efficacy of the process. In order to obtain optimal parameters for the removal of drugs with immobilized TiO_2_, the mutual influence of the initial concentration of the contaminant at environmentally relevant pH values was investigated using the response surface modeling approach. The faster kinetic rate of photocatalysis was obtained at pH 5 and at the smallest applied concentration of both drugs. The photocatalytic efficiency was mostly decreased by adding various inorganic salts and organic compounds to the drug mixture. Regarding the degradation mechanism of imatinib and crizotinib, hydroxyl radicals and singlet oxygen showed a major role in photochemical reactions. The formation of seven degradation products for imatinib and fifteen for crizotinib during the optimal photocatalytic process was monitored by high-resolution mass spectrometry (HPLC-QqTOF). Since the newly formed products may pose a hazard to the environment, their toxicity was studied using *Vibrio fischeri*, where the significant luminescence inhibition was assessed for the mixture of crizotinib degradants during the photocatalysis from 90 to 120 min.

## 1. Introduction

Drinking water supplies and water quality are increasingly affected by various micropollutants that were undetectable until recently, including pharmaceuticals as one of the most important new environmental contaminants. Modern technologies are enabling the increasing development and production of pharmaceuticals that serve to maintain quality of life [1,2,3]. Among the many classes of active pharmaceutical compounds, anticancer drugs have also received more attention and need to be monitored in the environment as their production and consumption increase, as confirmed by the steady increase in disease incidence; cancer is the second leading cause of death worldwide [4,5]. Their relatively good solubility, polarity, and increased mobility and persistence facilitate their entry into the environment through hospitals, improper disposal, household use, and ultimately through wastewater treatment plants that are not designed to remove small molecules such as pharmaceuticals [6,7]. The development of more sensitive analytical techniques over many years allows their detection at true µg/ng levels, which should not be neglected, as several anticancer drugs pose a direct threat to aquatic and terrestrial organisms due to their potential carcinogenic, mutagenic, genotoxic, and ecotoxic effects [2,6,8]. After release into the environment, the stability of the drug, which is usually excreted in a mixture with metabolites, is affected by various biotic and abiotic degradation processes, resulting in potentially more toxic degradation and transformation products [9,10,11].

For the successful protection of water sources and the possibility of their reuse, advanced oxidation processes such as photocatalysis are of great interest due to their wide application in the degradation of various organic molecules through the generation of non-selective and highly oxidative oxygen species [12,13,14]. TiO_2_ remains the most commonly used semiconductor material for photocatalytic reactions due to its good properties such as chemical and thermal stability, low cost, ease of applicability, and good photocatalytic efficiency in anatase crystal modification. The activation of TiO_2_ at a sufficient photon energy of the band gap between valence and conduction bands (3.2 eV) is possible by natural sunlight, UVA irradiance, which greatly reduces the energy consumption and harmfulness of commonly used UVC lamps [15,16,17]. One way to solve the problems with the separation of powder and water in suspension form and the reuse of the catalyst with the possibility of its dispersion in the environment is the sol-gel immobilization of TiO_2_ on a solid support, which ultimately allows easier and cheaper commercialization of the developed system [18,19]. The illuminated TiO_2_ forms electron-hole pairs that then reduce oxygen or oxidize water to produce reactive species responsible for pollutant degradation [20,21]. To evaluate the mechanism of heterogeneous photocatalysis, various organic and inorganic substances, called scavengers, can be added into an aqueous solution to interfere with reactive species involved in the degradation. Thus, alcohols such as isopropanol and tert-butanol are used to confirm the activity of hydroxyl radicals (·OH), p-benzoquinone scavenges superoxide ion radicals, azide reacts with singlet oxygen (^1^O_2_), while formic acid is used to determine the contribution of the holes in the valence band [22,23]. The Langmuir–Hinshelwood model is used to interpret the results of photocatalytic degradation kinetics, which is simplified at low analyte concentrations, so the pseudo-first model is often applied, as in this case (Equation (1)) [24]:(1)$$\frac{-dC}{dt}=k_{app}\cdot C\to C_{t}=C_{0}\cdot e^{-k_{app}t}\to ln\frac{C_{0}}{C_{t}}=k_{app}\cdot t$$
where *C*_0_ represents the initial concentration of the analyte, *C* is the concentration of the analyte at time *t*, and *k_app_* is the degradation rate constant. The half-life is expressed as: *t*_½_ = ln(2)/*k_app_*.

Two tyrosine kinase inhibitors, imatinib and crizotinib, used to treat leukemia and lung cancer, respectively [25,26,27], were the subject of this paper because they are potentially non-biodegradable, highly persistent compounds expected to be present in the aquatic environment [28,29]. Along with other anticancer drugs, these two pharmaceuticals cannot be removed from water by conventional water treatment methods [30], and so, photocatalysis was investigated to study the degradation kinetics under the mutual influence of contaminant concentration at environmentally relevant pH. Optimal process parameters were determined by applying response surface modeling (RSM) with Design-Expert software. TiO_2_ immobilized on a glass fiber mesh was used as the photocatalyst. A scavenger study was performed to gain better insight into the reaction mechanism. In addition, photocatalytic efficiency in drug removal can also be positively or negatively influenced by various organic and inorganic substances as natural constituents of water matrices. Thus, in this work, the influence of bicarbonate, phosphate, nitrate, and chloride ions as well as humic acids in an aqueous solution of pharmaceuticals was determined. High-resolution mass spectrometry (HPLC-QqTOF) was used to elucidate the preliminary structures of the degradation products formed under optimal photocatalytic conditions. Finally, the acute toxicity of pharmaceuticals and their mixture with degradation products was evaluated using marine bacteria *Vibrio fischeri* as test organisms. This study is expected to contribute to the existing knowledge on the degradation of antineoplastic pharmaceuticals.

## 2. Materials and Methods

### 2.1. Materials and Reagents

The anticancer drugs imatinib (IMT) and crizotinib (CRZ) at 99% purity were purchased from Pliva, Zagreb. Hydrochloric acid and sodium hydroxide, used for pH adjustment, were from VWR Chemicals, Radnor, PA, USA and Gram-mol, Zagreb, Croatia, respectively. HPLC grade acetonitrile, 2-propanol, and analytical grade disodium phosphate were purchased from Fisher Chemical, USA. Formic acid (98%) and sodium chloride (*p.a*.) were purchased from Lach-Ner d.o.o., Zagreb, Croatia. Ultrapure water was treated using a Millipore Simplicity UV system (Millipore Corporation, Billerica, MA, USA). Humic acids (HAs) were purchased from Sigma Aldrich, Switzerland. Sodium azide, sodium bicarbonate, and sodium nitrate (*p.a*.) were supplied by Kemika d.d., Croatia. TiO_2_ (Evonik, Aeroxide^®^, TiO_2_ P25 with a crystalline content of 75% anatase and 25% rutile) was used as the photocatalyst, immobilized on a glass fiber mesh adapted to the dimensions of the reactor using the sol-gel method described in a previous publication [31]. The average area of immobilized TiO_2_ on the solid support was 0.00356 ± 3.90 × 10^−4^ g/cm^2^ for IMT and 0.00373 ± 6.36 × 10^−5^ g/cm^2^ for CRZ.

The standard stock solution (1000 mg/L) of IMT was prepared by dissolving a precisely weighed amount of the powder in Milli-Q water, whereas CRZ was dissolved in acetonitrile. The organic solvent content in the CRZ working standard solutions was diluted to a negligible level (about 1%). The buffer solutions required for the hydrolysis experiments were prepared by mixing K_2_HPO_4_ and citric acid for pH 4; K_2_HPO_4_, NaOH, and water for pH 7; and NaOH, H_3_BO_3_, and water for pH 9 in a previously reported manner [32]. The standard solution of HAs was prepared at 50 mg/L with 5% (*v*/*v*) 1 M NH_4_OH and adjusted to pH 5.34 with 1 M formic acid. Working standard concentrations of HAs in the range of 5–20 mg/L were prepared by diluting the stock solution with Milli-Q water.

### 2.2. Photocatalytic Study

Photocatalytic experiments with two selected anticancer drugs were performed in an open rectangular reactor (dimensions: 17.5 × 4.6 × 3.3 cm) located in a dark room that allowed the exclusive use of the applied light module. The irradiation source consisted of two 6W ultraviolet lamps (I_0_ = 47.93 ± 0.26 W/m^2^, measured with a radiometer) with a broad maximum at 365 nm, placed above the reactor. The irradiation path was 2 cm. An amount of 100 cm^3^ of working standard solution was stirred with a continuous flow of 30 mL/min achieved by a peristaltic pump. The reactor was cooled to keep the temperature constant at 25 ± 1 °C during the experiments.

Before illumination, the pharmaceutical solution was contacted with the photocatalyst film on the glass fiber for 30 min to achieve adsorption/desorption equilibrium. The photocatalyst film was characterized and the results of SEM were presented in our previous paper [33], which were in agreement with the Raman spectroscopy measurements presented in this paper [34]. The sample after the dark sorption process was used as the initial concentration of photocatalysis. The degradation experiments were performed to investigate the dependence on the pH (5–9) and the initial concentration of micropollutants (5–15 mg/L). HCl and NaOH (0.01–0.1 M) were used to adjust the pH level of the solutions. After determining the optimal conditions for the photocatalytic degradation of IMT and CRZ using the RSM approach, a scavenger study was performed with the addition of isopropanol (·OH scavenger), formic acid (h^+^ scavenger), p-benzoquinone (O2·^−^scavenger), and azide (^1^O_2_ scavenger) to the drug solution to achieve a final concentration of 50 µM–50 mM for IMT and of 20 µM–20 mM for CRZ. The effects of different water constituents were studied by adding an appropriate aliquot of inorganic salts (NaNO_3_, NaCl, Na_2_HPO_4_, NaHCO_3_) and humic acid separately into the IMT/CRZ solution to obtain three concentration levels simulating the expected concentrations in real waters. In the experiments where the water matrix was studied, wastewater served as the diluent for the pharmaceutical solution instead of Milli-Q water. In all photocatalytic experiments, 500 µL samples were taken at specific time intervals. The degradation processes of IMT and CRZ were monitored using high-performance liquid chromatography with a diode-array detector (HPLC-DAD).

### 2.3. HPLC Analysis

Samples were measured at 258 nm for IMT, and 270 nm for CRZ, respectively, using HPLC-DAD, Agilent 1100 System (Santa Clara, CA, USA), with a C18 Kinetex column; 150 mm × 4.6 mm, 3.5 µm. The mobile phase consisted of 0.1% formic acid in Milli-Q water (A) and 0.1% formic acid in acetonitrile (B) in gradient mode. The gradient changed as follows: 0 min, 80% A; 6 min, 20% A; 7 min, 20% A; 7.01 min, 80%; 10 min, 80% A. The initial conditions were maintained for 4 min before the next injection. The flow rate was 0.5 mL/min with an injection volume of 25 µL. The limit of detection and limit of quantitation in ultrapure water were determined using the following equations: LOD = 3.3 s and LOQ = 10 s, where s denotes the standard deviation. The LOD and LOQ were 0.020 mg/L and 0.060 mg/L for IMT and 0.038 mg/L and 0.11 mg/L for CRZ, respectively.

### 2.4. Identification of Degradation Products

The degradation products formed during optimal photocatalysis were detected and identified using an Agilent 1290 Infinity II HPLC system in conjunction with an Agilent 6550 Series Accurate Mass Quadrupole (qTOF) mass spectrometer. A Zorbax Eclipse Plus C18 column measuring 2.5 mm × 50 mm, 1.8 µm at a temperature of 40 °C was used to separate the analytes. The mobile phase consisted of both 0.1% formic acid in water (solvent A) and acetonitrile (solvent B). The gradient of the mobile phase for IMT at a flow of 0.2 mL/min was as follows: 0 min 95% A, 0–6 min 60% A, 6–8.5 min 55% A, 8.5–15 min 5% A, 15–16.5 min 95% A. Crizotinib was eluted with the same flow but with a different gradient, which was as follows: 0 min 95% A, 0–4 min 85% A, 4–7.5 min 70% A, 7.5–8 min 65% A, 8–9.5 min 65% A, 9.5–12 min 80% A. The injection volume was 10 µL. The MS analyses were performed by electrospray ionization with the following parameters: sheath gas temperature 350 °C with a flow rate of 11 L/min, nebulizer pressure 35 psi, capillary voltage 3500 V, drying gas temperature 200 °C with a flow rate of 14 L/min. Nitrogen was used as a sheath and drying gas. HRMS spectra were performed in the range of 100–1000 *m/z*.

### 2.5. Toxicity Experiments

The measurement of the acute toxicity of IMT/CRZ and their mixtures with degradation products collected at different time intervals during optimal photocatalytic degradation was performed using a Hach Lange illuminometer LUMIStox 300 (Düsseldorf, Germany). Toxicity was determined by measuring luminescence inhibition before and after 30 min exposure of samples to marine bacteria *Vibrio fischeri* at 15 °C. Freeze-dried bacteria were used to prepare a fresh bacterial culture, which was then used for the measurements while the samples were diluted in 2% NaCl. The percentage of inhibition was calculated by comparing the luminescence of the sample with the luminescence of the control sample (NaCl).

## 3. Results

Prior to the photocatalytic degradation of pharmaceuticals, the hydrolytic stability of IMT and CRZ was determined according to the OECD 111 procedure [35]. Preliminary experiments included the preparation of IMT and CRZ standard solutions in buffer solutions at three pH values (pH 4; 7; 9) and the storage of the vials at 50 °C for a period of five days. Samples were analyzed using HPLC-DAD. The buffered IMT and CRZ samples were analyzed and compared with freshly prepared standard solutions, which were found to have less than 10% degradation at all pH values after five days (Figure 1). According to the OECD, these are hydrolytically stable compounds with an approximate half-life of more than 1 year at 25 °C [35].

### 3.1. Influence of Process Parameters: pH and Initial Concentration of Drug

To estimate the mutual effect of the two independent variables, pH and initial drug concentration, as photocatalytic influence parameters, the RSM approach was applied using Design-Expert software 7.0.0, Stat-Ease, Minneapolis, MN, USA. Since the obtained data fitted the first-order kinetics, the calculated *k*_app_ constants were used as the dependent variable of the model in response to the photocatalytic degradation induced by pH and concentration. The experimental data plan with the corresponding responses determined by a full factorial design at three levels (FFD) is shown in Table 1. Using multiple regression analysis, the influence of the process parameters was described by a quadratic polynomial equation, which is shown in Table 2 for each drug. The significance and accuracy of each model were evaluated by the statistical parameters (*p*, *R*^2^, *R*_adj_^2^), which were determined using the analysis of variance (ANOVA).

Both RSM models showed significance and accuracy, with *R*^2^ above 0.98 and *p* values less than 0.05 (Table 2). Thus, it can be seen that both pH and initial concentration affect the photocatalytic degradation of the studied drugs. According to the obtained kinetic rate constants, *k*_app_, the photocatalytic degradation of IMT and CRZ was slowed down by increasing the pH from 5 to 9. The pH is an important photocatalytic parameter that most affects the degradation efficiency, as it alters the differential charge of the photocatalyst surface and the degree of the ionization of the compound to be degraded, as well as the formation of reactive species involved in the degradation reactions [7,36,37]. The isoelectric point of TiO_2_ ranges from 6 to 6.5 [7,38,39], so the material can assume a positive or negative charge depending on the pH of the medium. Considering the *p*Ka values of IMT (7.8–8.07) and CRZ (5.6; 9.4) [3,40,41], anionic species of IMT are expected under alkaline conditions, while CRZ assumes a positive charge at acidic pH or a negative charge at basic pH. The photocatalysis of IMT and CRZ becomes less efficient with increasing pH, which may be due to the possible ·OH scavenging effect, repulsive forces in the saturated medium, etc. Secrétan et al. [3] also showed that the photocatalytic reaction for IMT is favored at acidic pH (5.4–6). In addition to the complexity of the above trend, previously published sorption experiments [33] showed higher sorption affinity of drugs on immobilized TiO_2_ at more alkaline conditions, which may explain the photocatalytic trend obtained. Stronger attractive forces under alkaline conditions promoted the greater sorption of drugs to TiO_2_, leading to a saturation of the adsorbent surface, which prevented the more successful photoactivation of the active sites predicted for radical formation, ultimately affecting the photocatalytic efficiency. The sorption effect may also be the cause of the best photocatalytic efficiency at the lowest initial contaminant concentration at all observed pH values. Imatinib as an amide and crizotinib as an amine are basic drugs that can react with H^+^ in acidic conditions, so it is not surprising that they have a similar tendency towards sorption and photocatalysis. According to the RSM models, pH 5 and an initial drug concentration of 5 mg/L were chosen as optimal for the rest of the experiments, which is consistent with similar results published for the photocatalytic degradation of ciprofloxacin [37]. The mutual interactions of the studied process parameters with k as the response were represented by 3D contour plots in Figure 2.

A higher pH of the solution can lead to a higher concentration of OH, which can have a negative effect on the degradation above a certain value due to the repulsive forces between the equally charged ions and the photocatalyst, which reduces the photoactivation of the surface [39]. The acceleration of degradation at the lowest initial concentration of the pharmaceuticals is due to the possible saturation of active sites on the catalyst as more moles of contaminant are introduced into the removal and the path length of photons entering the reaction system is reduced [37,42]. A similar trend for degradation can be assumed for real systems, following first-order kinetics, with much more dilute contaminant concentrations (which is more expected in the environment) [43].

It is evident that the concentration range of IMT and CRZ used is higher than the concentration commonly monitored in the environment (µg/ng scale), but thus it was possible to determine accurate residual concentrations of contaminants and elucidate the formation of degradation products by the analytical techniques used in this work.

### 3.2. Effect of Water Constituents on IMT and CRZ Degradation

Micropollutants enter real parts of the water system through different pathways, consisting of a large amount of different inorganic and organic species, thus affecting removal. To investigate the aforementioned effects on IMT and CRZ removal, each species normally present in three environmentally relevant concentrations in water was added to the photocatalytic reaction and described by the first-order kinetic model of degradation (Table 3).

#### 3.2.1. Nitrate Ions

The selected nitrate ion concentrations added to the aqueous solutions of drugs were 5, 25, and 50 mg/L to cover the range of concentrations present in the environment from 10^−5^ to 10^−3^ M [44]. Nitrates, which have a dual effect on pollutant degradation, can increase photocatalytic efficiency by promoting hydroxyl radicals [44,45] or inhibit the reactions as radical scavengers, according to Equations (2) and (3).
NO_3_^−^ + *h*^+^_vb_ → NO_3_·(2)
NO_3_^−^ + ·OH → NO_3_ + OH^−^(3)

In this study, a decrease in removal was observed with increasing nitrate concentration for both CRZ and IMT (Figure 3; Table 3), which is consistent with earlier reported results [45,46]. In addition to the scavenger effect, an inhibited reaction rate may also occur due to the occupation of active sites by nitrate ions instead of the contaminant. Both mechanisms of inhibition were confirmed by photodegradation experiments without photocatalysts (Appendix A), which showed an acceleration in the reaction rate upon the addition of nitrates to water (*k* = 0.0115 min^−1^ for IMT and 0.136 min^−1^ for CRZ) compared to direct photolysis (*k*= 0.0048 min^−1^ for IMT and 0.0113 min^−1^ for CRZ). A similar trend was previously observed in photolytic experiments by Ismail et al. [47], Xu et al. [44], and Dabić et al. [48].

#### 3.2.2. Bicarbonate Ions

The influence of HCO_3_^−^ as another ubiquitous component of natural water was determined by adding the anion to the reaction solution in the concentration range of 50–250 mg/L, which simulates real environmental conditions [46,49]. The degradation of CRZ was negatively affected by the bicarbonate scavenging effect of ·OH radicals, producing fewer reactive carbonate radical species (*k*(·CO3−) = 10^6^–10^7^ M^−1^ s^−1^; *k*(·OH) = 10^7^–10^10^ M^−1^ s^−1^) [50,51]. The pH of the pharmaceutical solutions was 8–8.5 for CRZ and 8.4–8.7 for IMT, which allowed the presence of most bicarbonate species in the aquatic solution (*pK*_a1_(H_2_CO_3_/HCO_3_^−^) = 6.37; *pK*_a2_(HCO_3_^−^/CO_3_^2−^) = 10.32) [47]. Thus, an increase in bicarbonate concentration and consequently increased alkalinity (more ·OH to scavenge) resulted in the prolonged degradation of both IMT and CRZ (Figure 4). However, the photocatalysis of IMT at lower HCO_3_^−^ concentrations showed a less significant effect on the pseudo-first-order rate. A slight increase in IMT degradation at 100 mg/L HCO_3_^−^ may be attributed to the compensation of newly formed carbonate radicals with high oxidative potential and the ability to reduce the recombination of electrons and holes by quenching e^−^ from the conduction band [43].

#### 3.2.3. Phosphate Ions

The presence of phosphate ions in the CRZ and IMT photocatalysis systems had a negative effect on the removal (Figure 5), confirming the hypothesis that phosphates are well adsorbed on immobilized TiO_2_ and affect the formation of active species such as h^+^ and ·OH, which are necessary for pharmaceutical degradation [43,47]. In contrast, previous publications showed an opposite sorption trend, where the negative charges of phosphates caused weaker adsorption on the catalyst and promoted degradation by the formation of more free ·OH radicals [52,53].

#### 3.2.4. Chloride Ions

The influence of chloride ions should also not be neglected, as CRZ and IMT are degraded more slowly in the presence of three different Cl^−^ concentrations (Figure 6). The photocatalytic inhibition can be attributed to the sorption of chlorides and the possible scavenging of h^+^ and ·OH [45,46], according to Equations (4)–(6).
Cl^−^ + *h*_vb_^+^ → Cl.(4)
Cl. + Cl^−^ → Cl_2_·−(5)
·OH + Cl^−^ → Cl.+ OH^−^(6)

#### 3.2.5. Humic Acids (HAs) as Dissolved Organic Matter

The results obtained (Figure 7) showed a negative influence on photocatalytic reactions of both IMT and CRZ, respectively, by the use of commercial humic acids as one of the main representatives of organic matter in natural water, usually at concentrations below 10 mg/L [54,55]. Inhibition in the presence of HAs may be caused by their radical scavenging activity, the occupation of active sites on the catalyst, and light attenuation effects. In addition to retarded photocatalysis, light absorption over a wide range of the UV spectrum can also stimulate the formation of oxidizing reactive species and promote pollutant removal in some cases [43,44,48]. The slowed kinetics can be attributed to the inner filter effect, i.e., the possibility of light absorption by HAs, not only by drugs represented by the overlap of the spectra of HAs and IMT/CRZ (Appendix A). However, it can be assumed that the slowed degradation of IMT and CRZ may be largely due to sorption on TiO_2_ and reactions with ·OH radicals, which is supported by the fact that HAs absorb UV light around 250 nm [56], while lamps with maximum emission at 365 nm were used in these experiments. An additional experiment with an HAs concentration of 5 mg/L was performed to see if the addition of isopropanol as a free radical scavenger of ·OH would further delay degradation. The obtained results confirmed the scavenging effect of HAs, as the degradation was slowed down (*k* = 0.0101 min^−1^ for IMT, *k* = 0.0126 min^−1^ for CRZ). The absorption spectrum of humic acids and drugs is shown in Appendix A.

Since the presence of pollutants in real water is affected by the mutual influence of all water constituents, two more photocatalytic experiments were performed with IMT/CRZ and a mixture of inorganic salts. The mixture consisted of 5 mg/L NO_3_^−^, 100 mg/L HCO_3_^−^, 10 mg/L HPO_4_^2−^, 100 mg/L Cl^−^, and 5 mg/L HAs. The results shown in Figure 8 confirmed that the synergistic effect of each species in the mixture slowed the degradation of both drugs (*k* = 0.0131 min^−1^ for IMT; *k* = 0.0281 min^−1^ for CRZ). Due to the competition of individual ions from the mixture for the active site on the photocatalyst and the scavenging effect, there was no drastic slowing of the reaction for IMT compared to the individual contribution of each species, whereas the removal of CRZ showed a stronger dependence on the water constituents. These experiments indicate that the removal of various organic pollutants is more complex in real water systems compared to the ideal (Milli-Q water). Therefore, another photocatalytic experiment was performed using a mixture of municipal and hospital wastewater (collected in October 2021 from the influent of a wastewater treatment plant as a sampling point in Italy; pH 6.88, conductivity 868 μS/cm, COD 182 mg/L O_2_, BOD 90 mg/L O_2_, DOC 18.5 mg/L, TSS 82 mg/L, N_tot_ 31 mg/L, P_tot_ 6 mg/L) as one of the main sources of anticancer drugs. The water sample was sterilized and filtered to avoid microbiological effects. A higher drug concentration (15 mg/L) was used due to the influence of the matrix and the presence of other contaminants that may affect the detection and quantification of the target compound. The photocatalytic removal efficiency of IMT and CRZ was lower (Figure 9), confirming once again the complexity of the different water constituents that affect the efficient removal of the analytes of interest through their indirect effect, as already described in this paper. The greatest contribution is expected from humic acids and phosphates, which normally have a good sorption capacity, causing the occupation of active sites on the photocatalyst the most. Besides the investigated inorganic and organic substances, other constituents such as pathogenic microorganisms, metals, and other organic/drug substances that are only partially removed in water treatment plants should not be ignored in wastewater. The simultaneous influence of pH also plays a major role because the water matrix increases the alkalinity of drug solutions and leads to reduced degradation. Specifically, the pH of the imatinib and crizotinib water matrix solution in pure water was above 7 (7.67; 7.72), while wastewater was more alkaline, with a pH above 8 (8.17; 8.26).

Table 4 shows the results obtained. The kinetics of the CRZ degradation in wastewater is more complex than that of IMT since it can be divided into two stages of first-order kinetics as a two-exponential function, where the first stage of the process was faster, with a *t*_1/2_ of 26.15 min, followed by the inhibition of the photocatalytic reaction, with a *t*_1/2_ of 256.72 min.

### 3.3. Scavenger Study

To better understand the photocatalytic mechanism and the role of reactive oxygen species (ROS) involved in the degradation of CRZ and IMT, different scavengers were used. Isopropanol (IPA) relatively selectively scavenges hydroxyl radicals, formic acid (FA)-positive holes, and p-benzoquinone (p-BQ)-superoxide anions and azide quenches singlet oxygen [15,22,57]. In Figure 10, it can be seen that both drugs are very susceptible to different ROS scavengers. The photocatalytic response was slowed most by the addition of IPA from 0.0168 to 0.0078 min^−1^ for IMT, suggesting that degradation is primarily driven by the formation of ·OH radicals. That they are not the only species extensively involved in IMT removal is shown by the addition of formic acid (46% of inhibition), which also demonstrates the activity of h^+^ that can affect other reactions in photocatalysis, such as the production of ·OH and singlet oxygen and the oxidation of the pollutant adsorbed on the catalyst [22]. The activity of O_2_·− and 1O_2_ was also confirmed, but with a significantly lower inhibition rate (13.7% and 21.4% respectively). In addition to the expected hydroxyl radicals, ^1^O_2_ shows the greatest responsibility for CRZ degradation (k from 0.138 to 0.0046 min^−1^), which could be explained by the formation of azide radicals (N_3_·) during the reaction with ·OH, which subsequently interferes with the substrate. Moreover, it has a tendency to form N_3_ as an electron donor with good sorption potential on TiO_2_ upon reaction with h^+^ [23,58].

### 3.4. Structure Elucidation of Degradation Products

The structures of the degradation products formed by photocatalysis of IMT and CRZ are tentatively shown in Figure 11 and Figure 13 with the proposed fragmentation pathways. Considering the fact that ·OH radicals are most responsible for IMT degradation, seven major degradation products can be formed as a result of OH-mediated oxidation, which include ·OH addition, H-abstraction, and electron transfer [3]. As previously published, six intermediates with the same m/z values were confirmed by Secrétan [3] and two by Wilczewska [59], but the localization of the hydroxyl groups in DP-3 and DP-6 was not the same. Non-selective ·OH radicals can react with different functional groups of the target molecule, making it difficult to determine a clear degradation pathway. Due to the reactivity of the amino group activating the aromatic ring [16], reactions at the N atom of the piperazine ring and the amino group of the phenyl ring have been proposed. Besides, IMT and CRZ are molecules with multiple aromatic rings such as pyridine, pyrimidine, benzene, and pyrazole, after which ·OH radicals show a higher affinity for attack compared to aliphatic molecules, but, in the case of IMT, the benzyl position was favorable, as in the case of electrochemical decomposition of IMT [60]. Another hydroxylated derivative with m/z 412 was also suggested as a possible degradation product. In the absence of pure DP standards, the results and the formation of degradation products were presented as the ratio of their relative area and the area of IMT or CRZ at the beginning of the experiment as a function of time (Figure 12 and Figure 14). The degradation products of IMT mostly reached their maximum concentration after 180 min (DP-3 and DP-7 after 30 min; DP-4 after 120 min) and five of them were completely degraded in the period of 360 min. The most abundant product was DP-6 with m/z 426, formed by the dealkylation of the N-C bond of the piperazine ring.

**Figure 11 nanomaterials-12-03532-f011:**
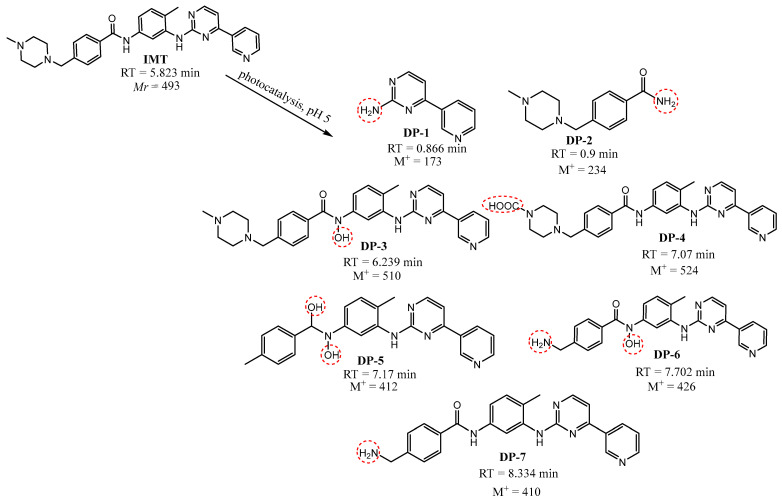
Proposed degradation products of IMT.

**Figure 12 nanomaterials-12-03532-f012:**
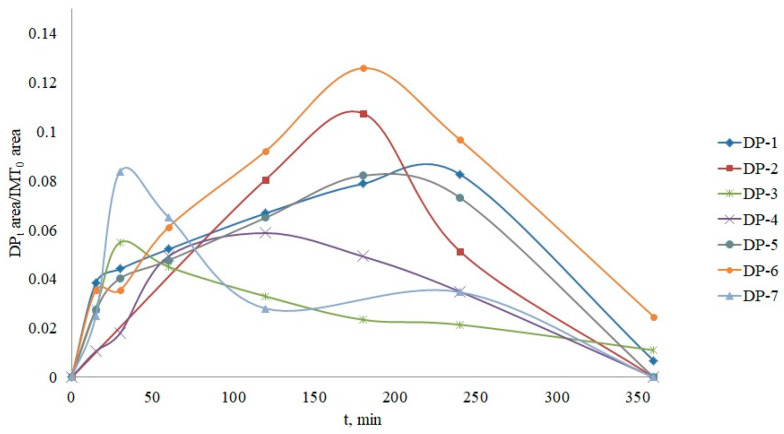
Formation and time profiles of imatinib DPs.

**Figure 13 nanomaterials-12-03532-f013:**
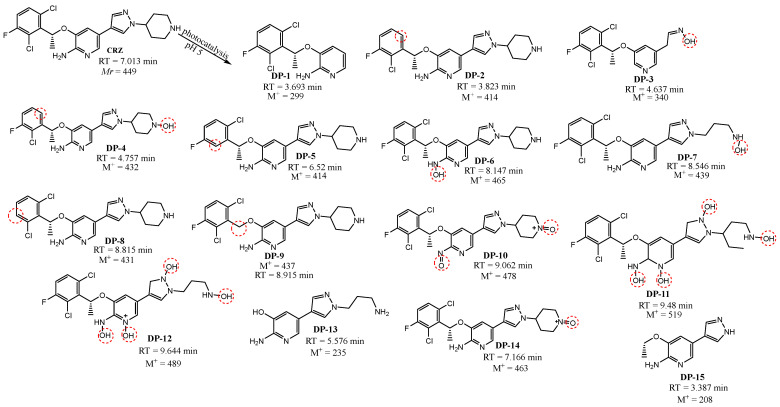
Proposed degradation products of CRZ.

**Figure 14 nanomaterials-12-03532-f014:**
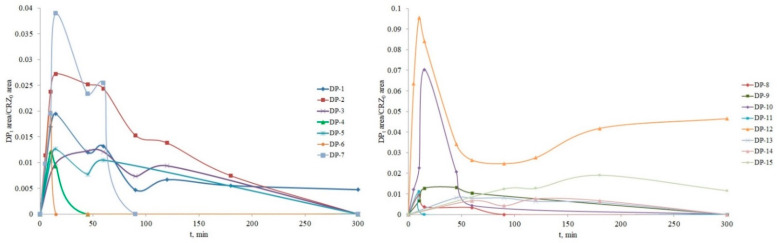
Formation and time profiles of crizotinib DPs.

During the photocatalysis of CRZ, fifteen possible degradation products were formed by oxidation and elimination reactions with ·OH radicals and singlet oxygen. Figure 13 shows the proposed structures of the degradation products, including the formation of hydroxylated intermediates, N-hydroxides and N-oxides, ring opening, N-dealkylation, dechlorination, and defluorination. In a previously conducted study on the forced degradation of CRZ [5], the formation of two DPs, corresponding to DP-6 and DP-10, formed under oxidative conditions by photocatalysis was described. Most DPs reached their maximum concentration after 10–15 min of photocatalysis, with their peaks mostly absent after 300 min of irradiation. The peaks with the highest intensity refer to DP-10, which is degraded after 60 min, and DP-12, which is the most abundant product. It stands out by a dynamic time profile that shows formation within five minutes of photocatalysis with a maximum in the tenth minute, followed by a decrease, and a re-increase in the concentration starting from the 90th minute. This result can be related to the toxicity results described in Section 3.5. The product ions for each proposed DP, obtained by MS/MS analysis, are listed in Appendix A for IMT and Appendix A for CRZ.

### 3.5. Toxicity Assessment

The ecotoxicological effects of anticancer drugs and the mixture of newly formed degradation products were evaluated by an acute toxicity test using the luminescent bacteria *V. fischeri*, which is commonly used to determine the negative contribution of micropollutants to the aquatic environment [2,6,44,61]. The percent of inhibition of bacterial luminescence when they are in contact with potentially toxic compounds is shown in Figure 15. The toxicity of IMT, which was completely degraded after 180 min, and the mixture of degradation products did not change over a given period of photocatalytic reaction; moreover, inhibition of less than 22% indicated a lack of acute toxicity. However, their long-term effects and potential antimicrobial activity in the presence of resistance should be further investigated. Toxicity for CRZ decreased in 45 min, while after complete degradation, the increase in inhibition rate was due to the production of DPs (for the period of 90 to 180 min I > 90%), which may pose a greater environmental hazard compared to the target compound. Specifically, the amounts of DP-1 and DP-12 increased again after 90 min of irradiation; DP-15 was detected in solution in 90 min, with maximum concentration in 180 min (Figure 14), suggesting that these products are the main protagonists of the negative effect on bacteria. The synergistic effect of the seven DPs present in the mixture between 90 and 180 min of photocatalysis may also lead to the greater inhibition of luminescence. Of course, these claims need to be further investigated by other tests, such as chronic toxicity tests conducted over a longer period of time with algae, fish, invertebrate test organisms, etc. [62,63], since it is known that drugs are continuously released into the environment [64]. The presence of IMT and CRZ in the aquatic environment should not be neglected, as compounds with an EC50 in the range of 10–100 mg/L can be classified as harmful to aquatic organisms, which was investigated in two other experiments using 100 mg/L as the nominal concentration of pharmaceuticals (EC50 = 63 mg/L for IMT; EC50 = 50 mg/L for CRZ) [65].

## 4. Conclusions

Although legislation is not yet in place, the problem of pharmaceuticals in the environment has been known for decades, and their diverse occurrence poses a long-term threat to drinking water. With the aim of improving knowledge on the removal of potential contaminants from the aquatic environment, two anticancer drugs, imatinib and crizotinib, were subjected to a photocatalytic process, investigating various process parameters that may influence the efficiency of the process. The results of photocatalysis were interpreted using first-order pseudokinetics. Considering the interactions between the structure of the molecules and the photocatalyst, favorable conditions were obtained for both pharmaceuticals in acidic conditions (pH 5) and at the lowest concentration tested (5 mg/L). The application of optimized initial process parameters revealed the importance of the influence of different water constituents on removal compared to ultrapure water, which delayed degradation more or less depending on the species. Although some of them can also have a positive effect on degradation, in this case, nitrates, phosphates, chlorides, and bicarbonates caused slower photocatalysis by competing with the pharmaceuticals for active sites on the catalyst and forming fewer reactive species. HAs, as a representative of the organic compound present in water, scavenge hydroxyl radicals and capture active sites on TiO_2_, slowing down the kinetics the most for both analytes. Experiments with a combination of all tested inorganic ions and HAs as well as with real wastewater confirmed the negative effect of removing the target substance, which is likely to be found in the complex matrix of the aquatic environment. The diversity of the water matrix, consisting of various ions, metals, and other pharmaceutic active ingredients, contributes to the slowing down of the photocatalytic reaction by competing with the catalyst, forming fewer reactive radicals and disabling the efficiency of the process in addition to increasing the pH and saturation of the solution and turbidity, which limits the penetration of light photons into TiO_2_. The scavenging experiments showed that IMT and CRZ are susceptible to other reactive oxygen species in addition to hydroxyl radicals, indicating the complexity of the reaction mechanism. The qualitative characterization of photocatalysis under optimal conditions was performed using HRMS, more specifically a Q-TOF mass spectrometer based on which all the degradation products formed were identified for both drugs tested. During the photocatalysis of IMT, six DPs were identified, while the degradation of crizotinib was accompanied by the formation of fifteen possible DPs. To our knowledge, this is the first report of crizotinib degradation in any form with a proposed degradation pathway. The formation of DPs was mainly by an oxidation mechanism. The toxicity assay with *Vibrio fischeri* showed a relatively low inhibition of less than 23% during the 360 min IMT photocatalysis, while the final stages of the CRZ degradation process indicated acute toxicity caused by the formation or increase in the concentration of degradation products. Although higher concentrations of pharmaceuticals are required for greater inhibition of bacterial cultures (IMT and CRZ, which are classified as environmentally harmful), their presence in the environment should not be neglected, so it is recommended to study the effects of long-term exposure to these substances.

## Figures and Tables

**Figure 1 nanomaterials-12-03532-f001:**
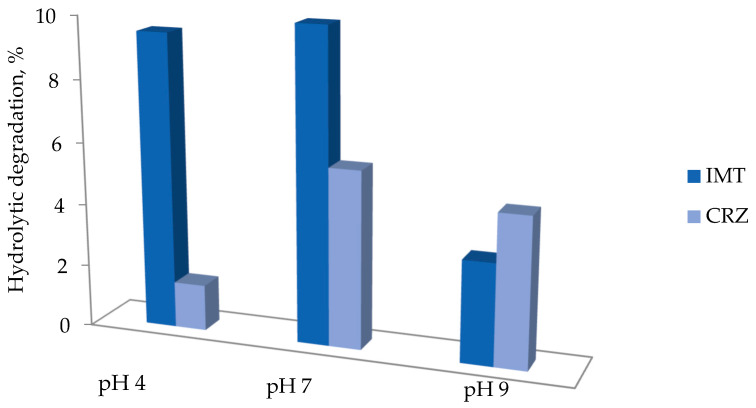
Hydrolytic stability of pharmaceuticals.

**Figure 2 nanomaterials-12-03532-f002:**
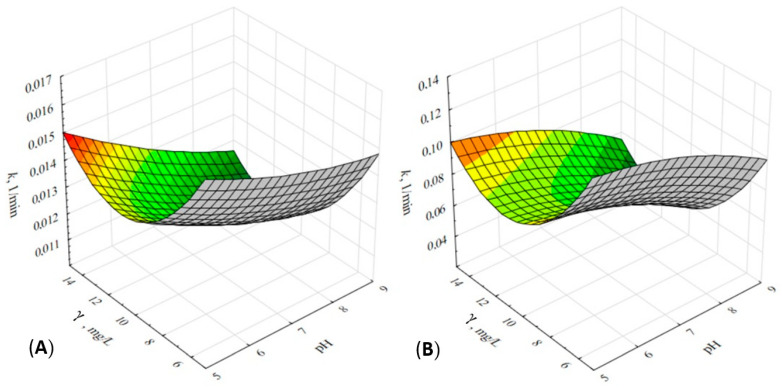
Mutual interactions between initial pH and the concentration of the pharmaceutical with k as the response, shown by 3D contour plots for IMT (**A**) and CRZ (**B**).

**Figure 3 nanomaterials-12-03532-f003:**
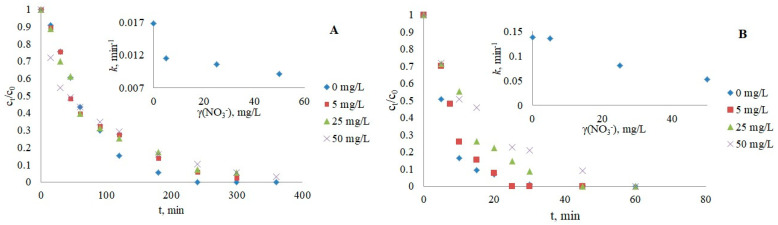
Influence of nitrates on the photocatalytic degradation of IMT (**A**) and CRZ (**B**).

**Figure 4 nanomaterials-12-03532-f004:**
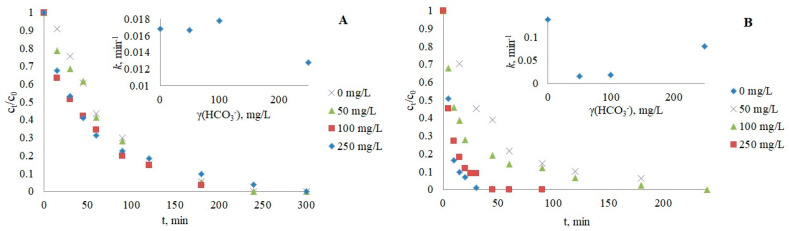
Influence of bicarbonates on the photocatalytic degradation of IMT (**A**) and CRZ (**B**).

**Figure 5 nanomaterials-12-03532-f005:**
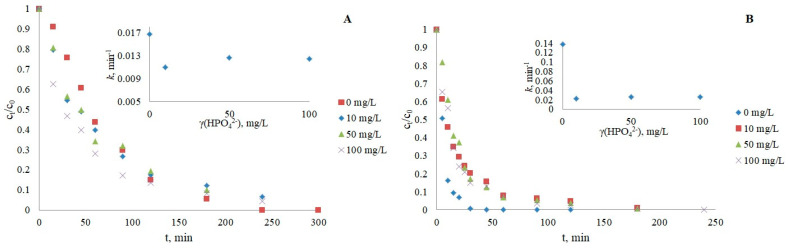
Influence of phosphates on the photocatalytic degradation of IMT (**A**) and CRZ (**B**).

**Figure 6 nanomaterials-12-03532-f006:**
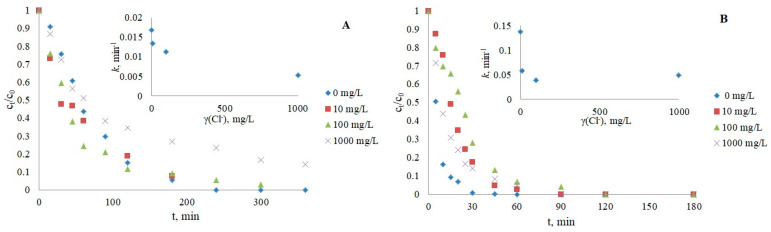
Influence of chloride ions on the photocatalytic degradation of IMT (**A**) and CRZ (**B**).

**Figure 7 nanomaterials-12-03532-f007:**
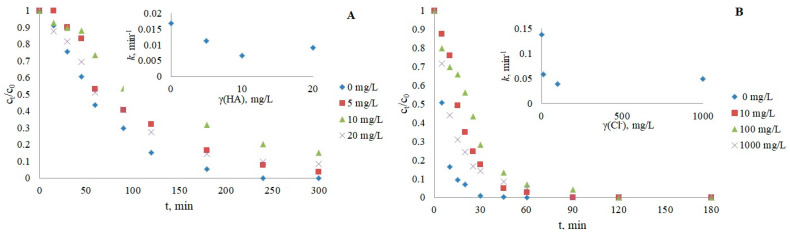
Influence of humic acids on the photocatalytic degradation of IMT (**A**) and CRZ (**B**).

**Figure 8 nanomaterials-12-03532-f008:**
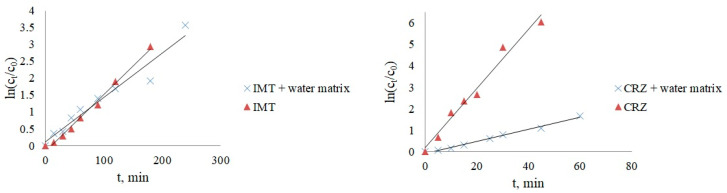
The effect of water matrix on IMT and CRZ removal.

**Figure 9 nanomaterials-12-03532-f009:**
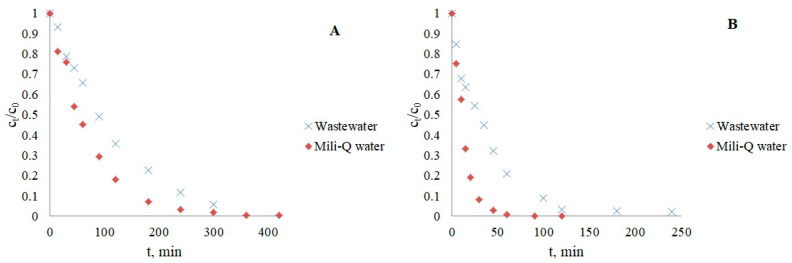
Kinetics of IMT (**A**) and CRZ (**B**) photocatalysis in wastewater.

**Figure 10 nanomaterials-12-03532-f010:**
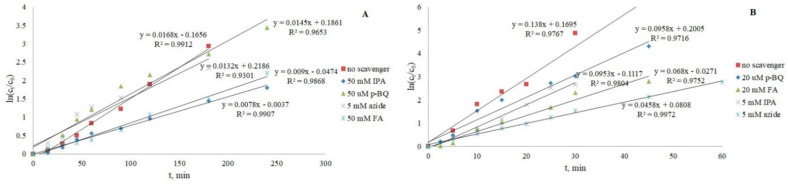
Identification of ROS active in IMT (**A**) and CRZ (**B**) removal.

**Figure 15 nanomaterials-12-03532-f015:**
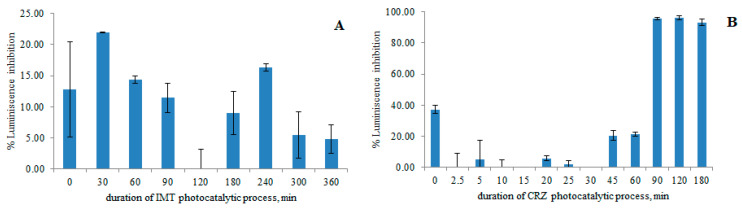
Change in the inhibition of luminescence during the photocatalysis of IMT (**A**) and CRZ (**B**).

**Table 1 nanomaterials-12-03532-t001:** Full factorial design matrix for the photocatalysis of IMT and CRZ with two independent variables expressed as real values and experimentally determined first-order degradation rates.

No.	Parameters		First Order Degradation Rate, *k* (min^−1^)	
	pH	*γ* (Pollutant), mg/L	IMT	CRZ
**1**	5	5	0.0168	0.1397
**2**	7	5	0.0154	0.1234
**3**	9	5	0.0148	0.0962
**4**	5	10	0.0137	0.0718
**5**	7	10	0.0115	0.0690
**6**	9	10	0.0111	0.0421
**7**	5	15	0.0148	0.1041
**8**	7	15	0.0135	0.0815
**9**	9	15	0.0113	0.0533

**Table 2 nanomaterials-12-03532-t002:** Equations and statistical data for RSM models developed for the IMT and CRZ photocatalytic degradation.

PhAc	Model Equation	Statistical Data			Influencing Model Factor (Based on *p*-Value)	Influencing Parameters
		*R* ^2^	*R* ^2^ _adj_	*p*		
**IMT**	*k* = 0.012 − 1.33·10^−3^A − 1.217·10^−3^B − 3.5·10^−4^AB + 3·10^−4^A^2^ + 2.35·10^−3^B^2^	0.9813	0.9517	0.0081	A, B, B^2^	pH, *γ*(IMT)
**CRZ**	*k* = 0.065 − 0.021A − 0.020B − 1.825·10^−3^ − 6.767·10^−3^A^2^ + 0.039B^2^	0.9840	0.9572	0.0068	A, B, B^2^	pH, *γ*(CRZ)

**Table 3 nanomaterials-12-03532-t003:** Effect of water constituents on the kinetics of photocatalysis of IMT and CRZ.

		IMT			CRZ		
Water Constituent	Concentration, mg/L	*k*, min^−1^	*R* ^2^	*t*_1/2_, min	*k*, min^−1^	*R* ^2^	*t*_1/2_, min
No addition	0	0.0168	0.9912	41.26	0.1380	0.9767	5.02
Nitrate	5 25 50	0.0115 0.0106 0.0091	0.9810 0.9768 0.9924	60.27 65.39 76.17	0.1361 0.0813 0.0523	0.9806 0.9848 0.9902	5.09 8.53 13.25
Bicarbonate	50 100 250	0.0167 0.0178 0.0128	0.9892 0.9827 0.9838	41.51 38.94 54.15	0.0157 0.0188 0.0809	0.9406 0.9408 0.9338	44.15 36.87 8.57
Chloride	10 100 1000	0.0134 0.0113 0.0053	0.9853 0.9473 0.9590	51.73 61.34 130.78	0.0584 0.0307 0.0496	0.9635 0.9654 0.9582	11.87 22.58 13.97
Phosphate	10 50 100	0.011 0.0126 0.0124	0.9753 0.9796 0.9617	63.01 55.01 55.90	0.0221 0.0255 0.0257	0.9346 0.9326 0.9330	31.36 27.18 26.97
Humic acids	5 10 20	0.0113 0.0066 0.0091	0.9927 0.9910 0.9728	61.34 105.02 76.17	0.0332 0.0099 0.0052	0.9509 0.9647 0.9484	20.88 70.01 133.30

**Table 4 nanomaterials-12-03532-t004:** Water matrix effect on IMT and CRZ photocatalysis.

	IMT			CRZ		
Experiment	*k*, min^−1^	*R* ^2^	*t*_1/2_, min	*k*, min^−1^	*R* ^2^	*t*_1/2_, min
PhAc + water matrix in Milli-Q	0.0131	0.9493	52.91	0.0281	0.9886	24.67
PhAc + wastewater	0.0094	0.9947	73.74	0.0265 0.0027	0.9863 0.9019	26.15 256.72

## Data Availability

Not applicable.

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
