# Peer review of "Photocatalytic Activity of TiO2 for the Degradation of Anticancer Drugs"

_nanomaterials, 2022, doi:10.3390/nano12193532_

Round 1
Reviewer 1 Report
The research article ‘Photocatalytic activity of TiO2 for the degradation of anticancer drugs’
In this research, the photocatalytic activity of TiO2 for the degradation of imatinib and crizotinib, which are anticancer drugs was studied. The Authors aim to show that TiO2 under UV illumination are able to degrade these drugs. Such parameters as inorganic ions, pH, drugs concentrations influence are deeply studied. Also, a deep analysis of degradation products and their toxicity is represented.
In general, TiO2 photocatalysis under UV illumination is not a novel topic as nowadays more attention is attracted to TiO2 and other nanostructures photocatalysis under visible light. However, this study is made correctly and deeply studied photocatalytic degradation of imatinib and crizotinib, and it makes this work different from others. Investigated influence of various parameters should attract the attention of the scientific community read this article.
The manuscript can be published after a minor revision and some improvements:
· Some analysis, such as SEM, XRD, XPS of used TiO2 immobilized on a glass fiber mesh before and after photocatalysis would greatly improve this manuscript.
· Images quality must be improved prior to publication. It is impossible to see data in some of them at the moment. In Fig 12, 14 graphs are not clear and dots should be made smaller to clearly see all data.
· The whole work is deep and many parameters and relations between them are studied. However, conclusions don’t show it. Conclusions should be rewritten to make them more specific and show many details are explained in a text before.
What is the structural difference between selected drugs? What process parameters were investigated? Why favourable conditions were obtained for both drugs at acidic conditions (pH 5)?
“Application of optimized initial process parameters revealed the importance of the influence of different water constituents on removal compared to ultrapure water, which delayed degradation more or less depending on the species.” – What constituents and how influenced photocatalysis? What species and how delayed degradation?
What main and most important degradation products were identified for both tested drugs?
“The toxicity assay with Vibrio fischeri showed a relatively low inhibition of less than 23 % during the 360 min IMT photocatalysis, while the final stages of the CRZ degradation process indicated acute toxicity caused by the formation or increase in the concentration of degradation products.” – How much toxic and which degradation products in your opinion are responsible for that?
Author Response
Some analysis, such as SEM, XRD, XPS of used TiO2 immobilized on a glass fiber mesh before and after photocatalysis would greatly improve this manuscript.
SEM analysis of immobilized TiO2 is previously published in the paper:
Tolić Čop, K., Mutavdžić Pavlović, D., Duić, K., Pranjić, M., Fereža, I., Jajčinović, I., Brnardić, I., Špada, V. Sorption Potential of Different Forms of TiO2 for the Removal of Two Anticancer Drugs from Water. Appl. Sci. 2022, 12, 4113.
The TiO2 preparation was the same, in the role of sorbent and also as photocatalyst, so we did not repeat the SEM images.
Also, in other previously published paper, a more detailed characterization of the TiO2 photocatalyst immobilized on glass mesh can be found, which includes additional SEM images, the Raman spectroscopy measurements, and difuse reflectance spectroscopy (DRS).
Malinowski, S.; Presečki, I.; Jajčinović, I.; Brnardić, I.; Mandić, V.; Grčić, I. Intensification of dihydroxybenzenes degradation over immobilized TiO2 based photocatalysts under simulated solar light. Appl. Sci. 2020, 10, 7571
Images quality must be improved prior to publication. It is impossible to see data in some of them at the moment. In Fig 12, 14 graphs are not clear and dots should be made smaller to clearly see all data.
Improved.
The whole work is deep and many parameters and relations between them are studied. However, conclusions don’t show it. Conclusions should be rewritten to make them more specific and show many details are explained in a text before.
Improved.
What is the structural difference between selected drugs? What process parameters were investigated? Why favourable conditions were obtained for both drugs at acidic conditions (pH 5)?
According to the structure of IMT and CRZ, they contain similar functional groups and molecules, so it is not surprising that they have a similar tendency to sorption and photocatalysis. According to IUPAC, imatinib as amide and crizotinib as amine indicate that they belong to basic pharmaceuticals. In their structure, they contain heterocyclic rings such as piperidine, piperazine, pyrimidine, etc., which contain a nitrogen atom that can act as an electron-donor and thus can be protonated. Thus, imatinib as an amide and crizotinib as an amine can react with H+ from acid under acidic conditions. Crizotinib is a zwitterion; it is protonated at a pyridine cation with a pKa of 5.6 and a piperidine cation with a pKa of 9.6, while imatinib has a pKa of about 8 (protonation of the piperazine nitrogen atom), which means that the molecules are protonated at pH 5. Their fate and tendency to degrade are also determined as a function of the surface properties of the photocatalyst, depending on the pH applied.
Application of optimized initial process parameters revealed the importance of the influence of different water constituents on removal compared to ultrapure water, which delayed degradation more or less depending on the species.” – What constituents and how influenced photocatalysis? What species and how delayed degradation?
Individual contributions of common natural constituents (chlorides, phosphates, bicarbonates, HA) in Milli-Q water that show a negative influence on photocatalytic degradation are identified in Section 3.2. The mutual influence of the mentioned substances has also shown the delay of the reaction. According to the obtained results, it can be said that there is also competition among the ions together with the analyte on the active sites of the photocatalyst, with the greatest contribution expected from humic acids and phosphates, which otherwise have a good sorption capacity. Depending on the type, wastewater also contains pathogenic microorganisms, metals such as copper, iron, lead, mercury, cyano compounds, living compounds and other pharmaceutical substances, in addition to the inorganic and organic substances, which are only removed to a limited and insufficient extent in the plants. Thus, in addition to the influence of sulfates, chlorides, organic matter, bicarbonates, etc., it is not possible to say exactly to what extent photocatalysis is slowed down in wastewater. Experiments on process optimization (section 3.1.) showed that degradation decreases with increasing alkalinity of the medium, which must also be taken into account. The water matrix affects the pH of the solution and consequently the ionization form of the drugs and all substances present. Thus, the pH of the imatinib and crizotinib water matrix solution in pure water was above 7 (7,67; 7,72) while wastewater was more alkaline with pH above 8 (8,17; 8,26).
What main and most important degradation products were identified for both tested drugs?
At this stage of the investigation, all degradation products are important in one way or another. To say something more precise about their importance, the DPs should be isolated and separated from the mixture to confirm the structure and test the influence of each DP. Although they were formed in lower amounts compared to the parent compound, the most stable ones with higher concentrations should probably be considered as the most important ones, since their more dominant role in this synergistic relationship can be assumed. Considering the time profiles of DP formation, it can be seen that DP-6 has the highest intensity and is not completely degraded in 360 minutes of UV/TiO2-IMT reaction, while DP-12, as the most abundant product formed within 5 minutes of CRZ photocatalysis, stands out by a dynamic time profile that shows a maximum in the tenth minute, followed by a decrease, and a re-increase of the concentration starting from 90th minute. This result can be related to the toxicity results.
“The toxicity assay with Vibrio fischeri showed a relatively low inhibition of less than 23 % during the 360 min IMT photocatalysis, while the final stages of the CRZ degradation process indicated acute toxicity caused by the formation or increase in the concentration of degradation products.” – How much toxic and which degradation products in your opinion are responsible for that?
Although the amount of product produced does not exceed 10% compared to crizotinib, considering the acute toxicity of the compound, a negative effect evident. Since the compounds were not isolated and the structures were not confirmed, the toxic effect is attributed to the synergistic effect of the mixture consisting of the few protagonists DP-1, DP-12 and DP-15. These are degradation products whose concentration increases after the 90th minute of reaction, while most other compounds are no longer detectable in the mixture.
Reviewer 2 Report
My comments are in attached file.

Author Response
In this study, the aim was to determine the photocatalytic activity of two anticancer drugs, imatinib and crizotinib, and to investigate various influences that may alter the kinetic degradation rate and ultimately the efficacy of the process.
Recorrected with Track Changes to “This study was aimed to determine the photocatalytic activity of two anticancer drugs, imatinib and crizotinib, and to investigate various influences that may alter the kinetic degradation rate and ultimately the efficacy of the process”.
The faster kinetic rate of photocatalysis was obtained at pH 5 and at the smallest applied concentration both drugs.
Recorrected with Track Changes to “The faster kinetic rate of photocatalysis was obtained at pH 5 and at the smallest applied concentration of both drugs”.
The photocatalytic efficiency was mostly decreased by adding various inorganic salts and organic compound in mixture with drugs.
Recorrected with Track Changes to “The photocatalytic efficiency was mostly decreased by adding various inorganic salts and organic compound in the drug mixture”.
Unclear sentence: “Experiments where the goal was to determine the presence of radical oxygen species, hydroxyl radicals and singlet oxygen showed major role in degradation of imatinib and crizotinib, respectively.”
Rewritten sentence.
“Regarding the degradation mechanism of imatinib and crizotinib, hydroxyl radicals and singlet oxygen showed major role in photochemical reactions.”
Drinking water supplies and water quality are increasingly affected by the presence of various classes of micropollutants that were undetectable until recently, including pharmaceuticals as one of the most important new environmental contaminants.
Recorrected with Track Changes to “Drinking water supplies and water quality are increasingly affected by various micropollutants that were undetectable until recently, including pharmaceuticals as one of the most important new environmental contaminants”.
Activation of TiO2 at sufficient energy of band gap between valence and conduction bands (3.2 eV) is possible by natural sunlight; UVA irradiance, which greatly reduces the energy consumption and harmfulness of commonly used UVC lamps [15-17].
Recorrected with Track Changes to “The activation of TiO2 at sufficient photon energy of band gap between valence and conduction bands (3.2 eV) is possible by natural sunlight, UVA irradiance, which greatly reduces the energy consumption and harmfulness of commonly used UVC lamps [15-17]”.
The illuminated TiO2 forms electron-hole pairs that then reduce oxygen or oxidize water respectively to produce reactive species responsible for degradation of the pollutant [20,21].
Recorrected with Track Changes to “The illuminated TiO2 forms electron-hole pairs that then reduce oxygen or oxidize water to produce reactive species responsible for the pollutant degradation [20,21]”.
“The Langmuir-Hinshelwood model is used to interpret the results of photocatalytic degradation kinetics, which is simplified at low substrate concentrations, so the pseudo-first model is often applied, as in this case (Equation 1.) [24].” What is a "substrate concentration"?
The term “substrate concentration” refers to the concentration of reactant/analyte.
where C0 represent initial concentration of the analyte, C is the concentration of the analyte at time t, and kapp is the degradation rate constant. The half-life was expressed as: t½ = ln(2)/kapp.
Recorrected with Track Changes to “where C0 represents the initial concentration of the analyte, C is the concentration of the analyte at time t, and kapp is the degradation rate constant. The half-life time was expressed as: t½ = ln(2)/kapp
Along with other anticancer drugs, these two pharmaceuticals cannot be removed from water by conventional water treatment methods [30], so photocatalysis was investigated to study the degradation kinetics under mutual influence of contaminant concentration at environmentally relevant pH.
Recorrected with Track Changes to “Along with other anticancer drugs, these two pharmaceuticals cannot be removed from water by conventional water treatment methods [30], and, so, photocatalysis was investigated to study the degradation kinetics under mutual influence of contaminant concentration at environmentally relevant pH”.
In addition, photocatalytic efficiency in drug removal of can also be positively or negatively influenced by various organic and inorganic substances as natural constituents of water matrices.
Recorrected with Track Changes to “In addition, photocatalytic efficiency in drug removal can also be positively or negatively influenced by various organic and inorganic substances as natural constituents of water matrices”.
High resolution mass spectrometry (HPLC-QqTOF) was used to elucidate the preliminary structures of the degradation products formed under optimal photocatalytic conditions.
Recorrected with Track Changes to “High resolution mass spectrometry (HPLC-QqTOF) was used to elucidate the preliminary structures of the degradation products formed under the optimal photocatalytic conditions”.
The anticancer drugs, imatinib (IMT) and crizotinib (CRZ) with 99 %purity of were purchased from Pliva, Zagreb.
Recorrected with Track Changes to “The anticancer drugs, imatinib (IMT) and crizotinib (CRZ) of 99 %purity were purchased from Pliva, Zagreb”.
Hydrochloric acid and sodium hydroxide, used for pH adjustment were from VWR Chemicals, USA and Gram-mol, Zagreb, respectively. HPLC grade acetonitrile, 2-propanol, and analytical grade disodium phosphate were purchased from Fisher Chemical, USA
Recorrected with Track Changes to “Hydrochloric acid and sodium hydroxide, used for pH adjustment, were from VWR Chemicals, USA and Gram-mol, Zagreb, respectively. HPLC grade acetonitrile, 2-propanol and analytical grade disodium phosphate were purchased from Fisher Chemical, USA”.
The buffer solutions required for the hydrolysis experiments were prepared by mixing K2HPO4 and citric acid for pH 4, K2HPO4, NaOH, and water for pH 7, and NaOH, H3BO3, and water for pH 9 in the manner previously described in the paper [32].
Recorrected with Track Changes to “The buffer solutions required for the hydrolysis experiments were prepared by mixing K2HPO4 and citric acid for pH 4, K2HPO4, NaOH and water for pH 7, and NaOH, H3BO3 and water for pH 9 in the previously reported manner [32]”.
Photocatalytic experiments with two selected anticancer drugs were performed in an open rectangular reactor (dimensions: 17.5 x 4.6 x 3.3 cm) located in a dark room that allowed the exclusive use of the light module applied.
Recorrected with Track Changes to “Photocatalytic experiments with two selected anticancer drugs were performed in an open rectangular reactor (dimensions: 17.5 x 4.6 x 3.3 cm3) located in a dark room that allowed the exclusive use of the applied light module”.
The degradation experiments were tested depending on the pH (5 – 9) and the initial concentration of micropollutants (5 – 15 mg/L) at different values, HCl and NaOH (0.01 –0.1 M) were used to adjust the pH of the solutions.
Recorrected with Track Changes to “The degradation experiments were carried out to see the dependence on the pH (5 – 9) value and the initial concentration of micropollutants (5 – 15 mg/L). HCl and NaOH (0.01 –0.1 M) were used to adjust the pH level of the solutions”.
In the experiments where the water matrix was studied, wastewater served as the diluent for the pharmaceutical solution, instead of Milli-Q water.
Recorrected with Track Changes to “In the experiments where the water matrix was studied, wastewater was served as the diluent for the pharmaceutical solution, instead of Milli-Q water”.
“The degradation processes of IMT and CRZ were monitored using HPLC-DAD.” Each abbreviation, including HPLC-DAD, should be defined on the first use.
Corrected to “The degradation processes of IMT and CRZ were monitored using high-performance liquid chromatography with a diode-array detector (HPLC-DAD)”.
Samples were detected at 258 nm for IMT, and 270 nm for CRZ, respectively using HPLC-DAD, Agilent 1100 System (Santa Clara, CA, USA) with C18 Kinetex column; 150 mm x 4.6 mm, 3.5 μm.
Recorrected with Track Changes to “Samples were measured at 258 nm for IMT, and 270 nm for CRZ, respectively, using HPLC-DAD, Agilent 1100 System (Santa Clara, CA, USA) with a C18 Kinetex column; 150 x 4.6 mm, 3.5 μm”.
Zorbax Eclipse Plus C18 column measuring 2.5 mm × 50 mm, 1.8 μm at a temperature of 40°C was used to separate the analytes.
Recorrected with Track Changes to “A Zorbax Eclipse Plus C18 column measuring 2.5 mm × 50 mm, 1.8 μm at a temperature of 40°C was used to separate the analytes”.
Acute toxicity of IMT/CRZ and their mixtures with degradation products collected at different time intervals during optimal photocatalytic degradation was performed using Hach Lange illuminometer LUMIStox 300 (Germany).
Recorrected with Track Changes to “Acute toxicity of IMT/CRZ and their mixtures with degradation products collected at different time intervals during optimal photocatalytic degradation was performed using a Hach Lange illuminometer LUMIStox 300 (Germany)”.
“To estimate the mutual effect of the two independent variables, pH and initial drug concentration as photocatalytic influence parameters, the RSM approach was applied using Design-Expert software.” Source of the software should be cited.
Corrected.
To estimate the mutual effect of the two independent variables, pH and initial drug concentration as photocatalytic influence parameters, the RSM approach was applied using Design-Expert software 7.0.0, Stat-Ease, USA.
“The significance and accuracy of each model were evaluated by the statistical parameters (p, R2, Radj2) which were determined using ANOVA.” ANOVA should be defined.
Corrected.
“The significance and accuracy of each model were evaluated by the statistical parameters (p, R2, Radj2) which were determined using the analysis of variance (ANOVA).”
Stronger attractive forces under alkaline conditions promoted greater sorption of drugs to TiO2, leading to saturation of the adsorbent surface, which prevented more successful photoactivation of the active sites predicted for radical formation, ultimately affecting the photocatalytic efficiency.
Recorrected with Track Changes to “Stronger attractive forces under alkaline conditions promoted greater sorption of drugs to TiO2, leading to a saturation of the adsorbent surface, which prevented more successful photoactivation of the active sites predicted for radical formation, ultimately affecting the photocatalytic efficiency.”
To investigate the aforementioned effects on IMT and CRZ removal, each species 267 normally present in three environmentally relevant concentrations in water was added to 268 the photocatalytic reaction and described by the first-order kinetic model of degradation 269 (Table 3).
Nitrates, which have a with dual effect on pollutant degradation, can increase photocatalytic efficiency by promoting hydroxyl radicals [43,44] or inhibit the reactions as radical scavengers according to Equations 2 and 3.
Recorrected with Track Changes to “Nitrates, which have a dual effect on pollutant degradation, can increase photocatalytic efficiency by promoting hydroxyl radicals [43,44] or inhibit the reactions as radical scavengers, according to Equations 2 and 3”.
In this study, a decrease in removal was observed with increasing nitrate concentration, for both, CRZ and IMT was observed (Figure 3; Table 3), which is consistent with previously published papers [44,45].
Recorrected with Track Changes to “In this study, a decrease in removal was observed with increasing nitrate concentration, for both, CRZ and IMT was observed (Figure 3; Table 3), which is consistent with earlier reported results [44,45]”.
A similar trend was previously published in photolytic experiments by Ismail et al. [46], Xu et al. [43], Dabić et al. [47].
Recorrected with Track Changes to “A similar trend was previously observed in photolytic experiments by Ismail et al. [46], Xu et al. [43], Dabić et al. [47].”
The presence of phosphate ions in the CRZ and IMT photocatalysis system had a negative effect on the removal (Figure 5), confirming the hypothesis that phosphates adsorb well on immobilized TiO2 and affect the formation of active species such as h+ and.OH, which are necessary for pharmaceutical degradation [42,46].
Recorrected with Track Changes to “The presence of phosphate ions in the CRZ and IMT photocatalysis systems had a negative effect on the removal (Figure 5), confirming the hypothesis that phosphates are well adsorbed on immobilized TiO2 and affect the formation of active species such as h+ and .OH, which are necessary for pharmaceutical degradation [42,46].”.
“Moreover, in previous publications, phosphates showed an opposite sorption trend, where adsorption negatively charges the catalyst and supports the formation of many more free .OH radicals [51,52].” Please correct this sentence for clarity.
Corrected.
“In contrast, previous publications showed an opposite sorption trend where phosphates negative charges caused weaker adsorption on the catalyst and promotion of degradation by formation of more free OH radicals [51, 52].”
The results obtained confirmed the scavenging effect of HA, as the degradation was slowed down (k = 0.0101 min-1 for IMT, k = 0.0126 min-1 for CRZ).
Recorrected with Track Changes to “The obtained results confirmed the scavenging effect of HA, as the degradation was slowed down (k = 0.0101 min-1 for IMT, k = 0.0126 min-1 for CRZ)”.
A higher drug concentration (15 mg/L) was used due to the influence of the matrix and the presence of other contaminants that may affect the detection and quantification of the target compounds.
Recorrected with Track Changes to “A higher drug concentration (15 mg/L) was used due to the influence of the matrix and the presence of other contaminants that may affect the detection and quantification of the target compound”.
Isopropanol (IPA) relatively selectively scavenges hydroxyl radicals, formic acid (FA) positive holes, p-benzoquinone (p-BQ) superoxide anions, and azide quenches singlet oxygen [15,22,56].
Recorrected with Track Changes to “Isopropanol (IPA) relatively selectively scavenges hydroxyl radicals, formic acid (FA) - positive holes, p-benzoquinone (p-BQ) - superoxide anions and azide quenches singlet oxygen [15,22,56]”.
Also IMT and CRZ are molecules with multiple aromatic rings such as pyridine, pyrimidine, benzene and pyrazole, after which .OH radicals show higher affinity for attack compared to aliphatic molecules, but in the case of IMT, the benzyl position was favourable, as in the case of electrochemical decomposition of IMT [59].
Recorrected with Track Changes to “Besides, IMT and CRZ are molecules with multiple aromatic rings such as pyridine, pyrimidine, benzene and pyrazole, after which .OH radicals show a higher affinity for attack compared to aliphatic molecules, but, in the case of IMT, the benzyl position was favourable, as in the case of electrochemical decomposition of IMT [59]”.
The peaks with the highest intensity refer to the DP-12, whose concentration increases after 90 minutes of reaction, and DP-10, which is degraded after 60 minutes.
Recorrected with Track Changes to “The peaks with the highest intensity refer to DP-12, which concentration increases after 90 minutes of reaction, and DP-10, which is degraded after 60 minutes”.
The product ions for each proposed DP, obtained by MS/MS analysis, are listed in Table S1 for IMT and Table S2 for CRZ, respectively.
Recorrected with Track Changes to “The product ions for each proposed DP, obtained by MS/MS analysis, are listed in Table S1 for IMT and Table S2 for CRZ”.
The percent of inhibition of bacterial luminescence when in contact with potentially toxic compounds was shown in Figure 15.
Recorrected with Track Changes to “The percent of inhibition of bacterial luminescence when they are in contact with potentially toxic compounds is shown in Figure 15”.
HA as a representative of the organic compound present in water slowed down the kinetics the most for both analytes.
Recorrected with Track Changes to “HA, as a representative of the organic compound present in water, slowed down the kinetics the most for both analytes”.